# Prescriptive Appropriateness: Inhospital Adherence to Proton Pump Inhibitors Deprescription Flow Chart

**DOI:** 10.3390/ph16050635

**Published:** 2023-04-22

**Authors:** Giammarco Baiardi, Giulia Calvini, Serena Panarello, Chiara Fioravanti, Manuela Stella, Antonietta Martelli, Giancarlo Antonucci, Francesca Mattioli

**Affiliations:** 1Clinical Pharmacology Unit, E.O. Ospedali Galliera, Mura delle Cappuccine 14, 16128 Genoa, Italy or giammarco.baiardi@edu.unige.it (G.B.); or manuela.stella@edu.unige.it (M.S.); 2Pharmacology and Toxicology Unit, Department of Internal Medicine, University of Genoa, Viale Benedetto XV 2, 16132 Genoa, Italy; 3Internal Medicine Unit, E.O. Ospedali Galliera, Mura delle Cappuccine 14, 16128 Genoa, Italy

**Keywords:** polytherapy, proton pump inhibitors, prescriptive appropriateness, deprescribing, patients’ clinical outcomes, clinical pharmacology, prescription drugs, drug interactions

## Abstract

The prescriptive appropriateness of Proton Pump Inhibitors (PPIs) in polypharmacy is controversial. PPIs are often overprescribed and the risk of prescribing errors and adverse drug reactions increases for each additional drug added to therapy. Hence, guided deprescription should be considered and easily implementable in ward practice. This observational prospective study evaluated the implementation of a validated PPIs deprescription flow chart to real-life internal ward activity through the presence of a clinical pharmacologist as an enhancing additional factor by assessment of inhospital prescriber’s adherence to the proposed flow chart. Patients’ demographics and prescribing trends of PPIs prescriptions were analyzed by descriptive statistics. The final analysis of data included ninety-eight patients (forty-nine male and forty-nine female), aging 75.6 ± 10.6 years; 55.1% of patients had home-PPIs prescriptions, while 44.9% received inhospital-PPIs prescriptions. Evaluation of prescriber’s adherence to the flow chart revealed that the percentage of patients with a prescriptive/deprescriptive pathway conforming to that of the flow chart was 70.4%, with low symptomatologic recurrences. The clinical pharmacologists’ presence and influence in ward activity may have contributed to this finding, since continuous training of the prescribing physicians is deemed a success-related factor in the deprescribing strategy. Multidisciplinary management of PPIs deprescription protocols shows high adherence by prescribers in real-life hospital settings and low recurrence events.

## 1. Introduction

Polypharmacy is defined by the use of at least five or more drugs taken daily by the same patient or by the continuative use of more medications than clinically indicated [1,2]. While on the one hand diagnostic accuracy, life expectancy and multimorbidities could in some cases justify the prescription of a multiplicity of medications, on the other hand being able to correctly follow the therapeutic regimen prescribed (therapeutic adherence) may be complex for some patients, exposing them to an increased risk of adverse drug reactions (ADRs) or a lack of therapeutic efficacy [3]. Polypharmacy, especially in the elderly patient, can also lead to reduced adherence to therapy, indeed, low adherence to chronic treatments increases with age from 56.6% in >65 years to 70.1% in ≥85 years [4]. Moreover, patients aged >65 years absorb more than 60% of the Italian National Health Service (NHS) expenditure [5]. Thus, in order to meet prescriptive appropriateness, it is not sufficient to abide by the SmPC’s indications; what the patient wants, the environmental context, and the social and family-related consequences of prescribing should be considered [6].

Prescribing a treatment following guidelines for specific diseases is certainly appropriate, but different literature sources caution overprescribing since prescribing errors appear to increase by 16% and ADRs by 13% for each additional drug added to the therapy [7,8]; thus, the prescribing physician should not forget to weigh how the new prescription may affect the whole therapy, paying particular attention to how each new drug may affect the benefit/risk ratio of the entire therapeutic regimen [9].

Proton pump inhibitors (PPIs) are among the most prescribed drugs in Italy, ranking among the top fifteen drugs in terms of consumption and among the top ten in terms of costs [10]. PPIs are available only as prescription medications and, with the exception of rabeprazole, are eligible for reimbursement by the Italian NHS. The different conditions for which these drugs may be prescribed, combined with the perceived perception that they are well tolerated, has made their use very common.

PPIs are irreversible inhibitors of gastric H^+^K^+^ATPase pumps [11,12]; they are mainly indicated for the treatment of gastroesophageal reflux disease (GERD), peptic ulcer disease, Helicobacter pylori (H. Pylori) infection and Zollinger–Ellison syndrome [11] and may be prescribed for prophylactic purposes to reduce possible injury from gastrolesive therapies such as nonsteroidal anti-inflammatory drugs (NSAIDs). Italian Medicines Agency’s AIFA has restricted its prescription through the NHS only, in the presence of some requirements indicated in two specific notes: note 1 and note 48 (see Materials and Methods section for details), issued precisely to reduce the inappropriate use of these therapies.

In a recent meta-analysis, the prevalence of GERD in adults in Europe was estimated to be 17.1% [13]; thus, the presence of GERD alone may partly explain why this class of drugs is widely used. The overprescription of PPIs could be due to defensive medicine purposes, in all those cases where a risk of gastrolesivity is suspected, even in the absence of associated risk factors, or when PPIs are requested by patients themselves.

The short-term use of PPIs is well tolerated and provides well known health benefits, although their prolonged use is associated with minor adverse health outcomes [14]. Many reviews of the scientific literature unequivocally demonstrate that PPIs are over-prescribed and misused contributing, often unnecessarily, to increasing the number of co-administered drugs [15,16]. Moreover, PPIs have been shown to impair the pharmacokinetic profile of relevant drugs, raising concerns on their potential clinical impact in terms of drug–drug interactions (DDIs) when used concomitantly [17]. Although PPIs are often considered well-tolerated with modest side effects, unfortunately many studies have documented probable causal links between the continuous use of PPIs and serious adverse events (SAEs), such as Clostridium difficile infections [18,19], fractures [20], hypomagnesemia [21], acute and chronic kidney disease [22], incremented cardiovascular risk in patients under anti-aggregating therapy [23], vitamin B12 deficiency [24] and a possible association with dementia [25] or gastrointestinal carcinoids [26].

The evidence of these possible adverse events (AEs), coupled with a higher probability of lethal events (about 20%) among PPIs users even in those who do not have gastrointestinal pathologies [27], shows how the prescriptive appropriateness of these drugs must be carefully re-evaluated. An Italian hospital-based study found that more than 62.4% of inpatients receiving PPIs at hospital admission, were treated out of indications and this percentage rose to 63.2% at discharge [28].

Therefore, it is necessary to provide tools which can support the prescribing decisions or when a guided deprescription strategy should be considered. Credit must be given to Barbara Farrell and colleagues from the University of Ottawa for developing the first tool for the PPIs deprescription, specifically designed to guide physicians in their choice in line with therapeutic appropriateness [29]. We therefore conducted a prospective observational study applying the deprescriptive flow chart of Farrell et al. to patients admitted to the Department of Internal Medicine of the Galliera Hospital in Genoa (Italy), evaluating its effect on the prescription of PPIs at discharge from the hospital.

This paper focuses on the deprescriptive flow chart application during three months of ward activities, in particular evaluating the applicability to hospital clinical practice of the PPIs prescription/deprescription flow chart and its adherence abided by hospital prescribers, with the presence of a clinical pharmacologist.

## 2. Results

Over the three month study period one hundred and twelve patients (fifty-five female and fifty-seven male) were evaluated; drop out involved fourteen patients (three deceased and eleven transferred to another ward).

Final analysis of data included ninety-eight patients (forty-nine male and forty-nine female), whose mean (±standard deviation, SD) age was 75.6 ± 10.6 years (female 77.2 ± 12.0 and male 74.1 ± 8.9); 79.6% were on polypharmacy (number of drugs ≥5), among them about one-third (33.3%) were exposed to more than ten concurrent treatments.

Fifty-five percent (55.1%) of patients had PPIs prescribed before entering the ward (home-PPI), while for 44.9% of them PPIs prescriptions occurred during hospitalization (inhospital-PPI). Pantoprazole at a dose of 40 mg once daily (PAN40) was the most prescribed PPI (74%). PPIs’ full doses [PAN40 or lansoprazole 30 mg (LAN30), once daily] were taken by 83.3% of patients who had home-PPIs prescriptions and 93.2% treated with inhospital-PPIs. Table 1 shows patients’ demographics and prescribing trends.

Stratification of eligible patients by the six possible clinical prescribing scenarios (defined in Methods from A to F) is shown in Figure 1. In whole population analysis, PPIs prescription/deprescriptions followed the proposed flow chart in 70.4% of cases, while 29.6% were done off the flow chart.

Of the fifty-four patients who had home-PPIs prescriptions (Figure 1), forty (74.1%) were discharged according to flow chart prescriptive indications. In thirty patients (75.0%), PPIs therapy was confirmed at discharge as complying with recommendations (clinical scenario A), in ten patients (25.0%) it was correctly discontinued at discharge (clinical scenario C) and in no case was the dosage scaled at discharge (clinical scenario B).

Of the forty-four patients who had received inhospital-PPIs (Figure 1), twenty-nine (65.9%) were discharged according to the flow chart prescriptive indications, in twenty-one patients (69.0%) the therapy was confirmed at discharge (clinical scenario D), while in six cases (20.7%) it was not confirmed at discharge (clinical scenario F) and in two patients (6.9%) the dose of PPIs prescribed during hospitalization was reduced upon discharge (clinical scenario E).

Of the twenty-nine patients for whom PPIs prescriptions/deprescriptions were off the flow chart (Figure 1), fourteen home-PPIs patients were not deprescribed despite the fact that the drug could be potentially discontinued and fifteen received inhospital-PPIs prescriptions even though not indicated by the flow chart. All twenty-nine patients followed by clinicians off the flow chart were discharged with PPIs.

Form B (see Table 3) analysis of the fourteen home-PPIs patients with non-deprescriptions despite possible PPIs discontinuation showed that: in five patients it was not possible to trace the reasons for non-deprescriptions and the prescriptions were therefore categorized as improper; two patients had only prior symptoms; in two patients the reason for non-deprescription was justified, in form B, by virtue of the generic “frailty” status of the patients; one patient was not deprescribed due to concomitant corticosteroid intake; one patient due to polypharmacotherapy (ten drugs co-administered); two patients due to anemia, and finally, one patient with chronic renal failure (CRI, grade III), who was taking it by nephrological prescription for the management of uremic gastritis was not deprescribed.

Form B analysis of the fifteen inhospital-PPIs patients with out-of-flow chart new prescriptions showed that: in three cases it was not possible to trace the prescriptive reasons and the prescriptions were therefore categorized as improper;in another five cases the prescription was justified by prescribing physicians for the presence of high-dose corticosteroids in therapy; five patients were prescribed PPIs because they were considered “frail” (of these five, two had anemia under investigation); finally, two subjects received “precautionary” prescriptions of PPIs because they were polytreated (one patient with ten drugs and one patient with eight drugs, respectively).

## 3. Discussion

Elaborating and implementing deprescription protocols both in the context of hospitals and territorial healthcare entails collaboration among different professions in a multidisciplinary management perspective. This management system is useful not only for patient care, but more in general to identify clinical best practices to be put in place in different care contexts. The role of clinical pharmacologists becomes crucial both in identifying the molecules that are more subject to poor prescriptive adequacy, and distinguishing possible interaction profiles. Furthermore, in a nosocomial context, this role can function as a connection between hospital pharmacy and prescribing physicians in the search for a balance between economic, managerial and clinical requirements.

Specific prescription rationalization programs can prove to be useful and effective tools for reducing the consumption of drugs that are widely used, often without being indicated. It is worth noting that hospital applications of deprescription protocols, however important, cannot replace the far more difficult task of transferring a more judicious prescriptive thought model to territorial healthcare. Naturally hospitals are very different to territorial healthcare not only because all of the prescribed drugs are actually administered, but also because physicians work in smaller, controlled environments. In particular the clinical conditions of hospitalized patients are certainly more severe, and the size of the hospital population is much smaller than that of a whole area of interest of territorial healthcare.

In the introduction we explored how the initial choice of a category of drugs such as PPIs was not random, but was linked to their extensive and often inconsiderate use both due to the frequency of symptoms they address, and the mistaken perception that they are risk-free drugs. PPI misuse, thus without a clear approved indication, is estimated to be between 20% to 80% worldwide [29,30,31]. In the hospital setting the percentage of unnecessary PPIs prescriptions started at admission and continued at discharge is about 70% [32,33,34]. The absence of a periodic medical review of these treatments is a leading cause of their long-term continued inappropriate use after discharge [15,30] and rebound symptoms due to PPI-induced hypergastrinemia are possibly driving the cycle of inappropriate prescribing [35]. Tremendous cost in financial terms is consequently reported in many Western countries [36]. Inhospital implementation of successful PPI deprescription strategies are therefore issued. Concerns on PPIs safety on their long-term use is increasing despite being well tolerated and efficacious in the short-term [37]. The most important success-related factors appear to be the training of the physicians responsible for deprescribing and the clarity and the simplicity of the de-escalation protocol in guiding patients in the event of recurrence of symptoms [38].

There is no uniform consensus on PPIs deprescription, though consideration of benefits/risk assessment of therapeutic indication should be involved in the process of reducing and/or stopping the PPIs therapy. Different deprescribing approaches have been previously tested and evaluated [36]. The present study approach focuses on the Canadian algorithm [29] implementation in a real-life internal ward activity through the presence of a clinical pharmacologist as an enhancing additional factor. The ultimate goal of deprescribing is to limit polypharmacy, which exposes patients to possible ADRs and DDIs.

It seems clear that in the setting in which we operated, the general trend toward the use of polypharmacy was confirmed; 79.6% of inpatients were treated with more than five drugs daily. Evaluation of inhospital prescriber’s adherence to the proposed flow chart revealed that the percentage of patients with a prescriptive/deprescriptive pathway conforming to that of the flow chart was 70.4%.

The clinical pharmacologists’ presence and influence in the ward activity may have contributed to this finding of high adherence, since continuous training of the physicians responsible for deprescribing is deemed a success-related factor in the deprescribing strategy [38]. In our opinion, it supports the thesis according to which multidisciplinarity is the keystone for good management of therapies, in particular in the current health context in which there is an increasingly higher number of available drugs and a growing number of polypathological and poly-treated patients.

Despite the possibility of counselling on therapy management, a significant 29.6% of prescribers’ decisions deviated from the flow chart. The most frequent causes of inappropriate prescription or non-deprescription were prophylaxis in patients taking only corticosteroids or anticoagulants with no other additional risk factor or generic gastroprotection in polypharmacy. Contrary to widespread belief, corticosteroids and anticoagulants do not cause any direct injury to the gastrointestinal mucosa, thus PPI co-therapy is not routinely indicated [39]. Moreover, polypharmacy expresses a bidirectional relationship with frailty, but could be a major contributor in its development [40]. Unless clinically necessary, the introduction of even one additional and reputedly safe drug (i.e., PPI) must be carefully evaluated in complex therapies. In this context polypharmacotherapy could actually be in itself a fragilizing factor for the patient, both in terms of interactions and treatment adherence.

Despite prescribers’ high adherence to the proposed deprescription algorithm, one of its main limitations was the application to the acute cases encountered for which it was not possible for clinicians to always adhere to the provided suggestions. In this respect, it is not conceivable not to align with the position of prescribing colleagues in deviating from the flow chart. However, early detection of symptoms recurrences during hospitalization allows the prompt reintroduction of PPI therapy. Consequently, only for few patients (eight) the self-filling GERD Impact Scale (GIS) questionnaire showed a score <3 (presence of uncontrolled symptoms). PPI treatment was then reintroduced and maintained at discharge. An advantage of this study’s design was hence the possibility of being able to directly follow the symptomatologic consequences of the prescribing-deprescribing decision over the course of hospitalization.

Medicine is changing rapidly and is increasingly headed towards so-called “target therapies”. In a context that brings us to increasingly focus on pathologies rather than patients, it is important to remember that pharmacological treatment does not act only on the illness, it also acts on the individual.

Clinical pharmacologists enter this delicate balance bringing, on the one hand, greater competence in purely pharmacological terms regarding the principles of pharmodynamics and pharmacokinetics which regulate the functioning of drugs and their mutual influence. On the other, they focus attention on the patient with a careful evaluation of risk/benefit, adverse effects and tolerability. The quality of life that can be guaranteed to patients with available therapies is the primum movens of our activities as clinicians, net of available evidence and scrupulous collegial evaluation, which is the goal of our work. While we understand the difficulty of deprescribing a drug chronically taken by a comorbid and poly-treated patient in an acute care hospital setting, it seems necessary to implement all possible strategies to reduce the medication burden in the hospital/territory transition.

The problem of polypharmacy is well known, as is its high cost in terms of adherence, patient quality of life and expense for the National Health System. An interdisciplinary approach with the support of a clinical pharmacologist together with deprescribing tools like the flow chart used in this study could be a valid vehicle for rationalizing prescriptions.

## 4. Materials and Methods

The primary objective of the present study is to evaluate the applicability of a validated PPIs deprescription flow chart in a hospital setting by the assessment of an inhospital prescriber’s adherence to the proposed flow chart in the presence of a clinical pharmacologist as an enhancing factor. The primary endpoint is the percentage of patients with a prescriptive/deprescriptive pathway conforming to that of the flow chart. The study, approved by the Regional Ethics Committee (Authorization nr. 032/2019), did not provide for the execution of any invasive method or variation of the normal ward clinical practice.

The study flow chart used was an adaptation of B. Farrell et al. validated in Italian by V. Maio (PharmD, MS, MSPH, College of Population Health, Thomas Jefferson University, USA) and S. Del Canale (MD, Local Health Authority of Parma, Italy) and slightly modified to abide by Italian Medicines Agency’s (AIFA) Notes 1 and 48, which limit the PPIs prescriptions necessary for full reimbursement by the National Health Service (NHS) [41,42]. In detail, Note 1 provides for NHS reimbursability of PPIs for the prevention of serious upper gastrointestinal tract complications in patients chronically treated with non-steroidal anti-inflammatory drugs (NSAIDs) or on antiplatelet therapy with low-dose acetylsalicylic acid (≤100 mg daily), provided that one of the following risk conditions is met: (i) history of previous digestive haemorrhage or peptic ulcer not healed by eradication therapy, (ii) concomitant anticoagulant/cortisonic therapy, and (iii) advanced age [41].

Note 48, on the other hand, provides for reimbursability through the NHS of PPIs only for specific clinical conditions and establishes the maximum treatment duration. PPIs are reimbursable for a 4 week treatment (occasionally 6 weeks) in the following cases: (i) duodenal or gastric ulcer positive for H. pylori, (ii) for the first week, or the first two weeks, in combination with infection eradicating drugs, (iii) first episode of H. pylori-negative duodenal or gastric ulcer or (iv) first episode of gastroesophageal reflux disease with or without esophagitis. Note 48 also provides for NHS reimbursement of PPIs for prolonged treatment in cases of Zollinger–Ellison syndrome or relapsing cases of duodenal or gastric H. pylori-negative ulcer and gastro-esophageal reflux disease with or without esophagitis. In these cases, re-evaluation of therapy should be carried out after one year of treatment [42].

The study design had two distinct phases. An initial training period, during which the objectives and design were explained and discussed with the multidisciplinary team. The team consisted of: internists and clinical pharmacologists, nurses and pharmacists. The training period was followed by a 3 month implementation phase. During this second phase, ward physicians were required to apply the flow-chart for prescribing/deprescribing PPIs to all enrollees.

All patients admitted to the Department of Internal Medicine having age ≥18 years and having a PPI already in therapy at the time of admission to the department and all patients aged ≥18 years to whom a PPI was being added during hospitalization were included. Patients with severe neurological/psychiatric problems, advanced cancer disease, or having a life expectancy of less than 12 months were excluded. Patients affected by any medical condition which did not allow full understanding and the ability to fill in the study’s provided self-filling GERD Impact Scale (GIS) questionnaire [43], were also excluded.

The following patient’s clinical and demographic characteristics were collected in an appropriate form (form A, not included): age, gender, date and reason for admission, time of PPI prescription (before or during hospitalization), prescriptions’ strength (i.e., the amount of the drug and the dosage formulation for each PPIs prescription), pathologies which could justify the prescription of a PPI, concomitant therapies, diagnosis and date of discharge and the consequent prescription/deprescription of the PPI. Form A had to be completed in all its parts by the ward clinical pharmacologist, by the date of discharge.

The eligible patient was prescribed/deprescribed a PPI following the proposed flow chart or according to the clinical needs. The resulting 6 prescriptive scenarios are shown in Table 2.

**Table 2 pharmaceuticals-16-00635-t002:** Possible prescriptive scenarios.

*(A)* *PPI upon admission → continuation of PPI upon discharge;* *(B)* *PPI upon admission → change in PPI dose (increase/reduction) upon discharge;* *(C)* *PPI upon admission → complete deprescription of PPI upon discharge;* *(D)* *ex novo prescription of a PPI during hospitalization → continuation of the PPI upon discharge;* *(E)* *ex novo prescription of a PPI during hospitalization → change in PPI dose (increase/reduction) upon discharge;* *(F)* *ex novo prescription of a PPI during hospitalization → complete deprescription of PPI upon discharge.*

If, for any reason, the internal medicine clinician deviated from the indication of the flow chart, she/he was asked to fill in a second form (Form B, Table 3) to record inadherence rationale, for the flow chart represented a mere therapeutic suggestion that could be accepted or ignored by the care provider. Form B consisted of 8 free-response items that addressed the reasons leading the physician to prescribe/deprescribe a PPI or to change the PPI dose (increase/reduction) not recommended in the flow chart; the reasons, on the other hand, that led to the failure of deprescription even though recommended by the flow chart and, finally the main critical issues encountered in the interpretation and/or implementation of the prescriptive flow chart.

**Table 3 pharmaceuticals-16-00635-t003:** Form B, inadherence rationale.

Ex novo prescription of a PPI not recommended in the flow chart? ✓ Reasons for prescription (specify symptomatology and clinical benefit) Failure to de-prescribe PPI therapy (only where deprescription is recommended by flow chart)? ✓ Reasons for choice (specify symptomatology and clinical benefit) Dose increase (only in case where increase is not recommended by flow-chart)? ✓ Reasons for choice (specify symptomatology and clinical benefit) Dose reduction (only in case when not recommended by flow-chart)? ✓ Reasons for choice (specify symptomatology and clinical benefit) Complete deprescription of PPI (only in the case when not recommended by flow-chart)? ✓ Reasons for choice (specify symptomatology and clinical benefit) Change in active ingredient/mode of administration (only in case not recommended by flow-chart)? ✓ Reasons for choice (specify symptomatology and clinical benefit) Other (any other modification of PPI therapy performed without following the reference flow-chart): ✓ Reasons for choice (specify symptomatology and clinical benefit) Main critical issues encountered in the interpretation and/or implementation of the prescriptive flow-chart?

Moreover, to assess gastrointestinal symptoms the present study planned to administer to the patients the GIS questionnaire validated in Italian [43] and formulated in a patient-friendly language. In addition to the actual symptom area the GIS questionnaire takes into consideration GERD’s effects in disturbing the normal course of daily patient’s activities and their impact on quality of life, in compliance with Montreal’s criteria [44]. The GIS questionnaire was submitted to all patients at the time of enrolment and repeated only in patients showing recurrence of symptoms.

Relevant data were retrieved from Form A and B and categorized anonymously in an appropriate database; descriptive statistics were then undertaken to summarize results.

## 5. Conclusions

Multidisciplinary management of PPIs deprescription protocols shows high adherence by prescribers in real-life hospital settings and low recurrence events. The rational appropriate use of drugs, according to indication and dosage, and mediating between scientific evidence, the actual clinical need and patient needs, should be the best therapeutic strategy both in terms of health economics and quality of patient life.

## Figures and Tables

**Figure 1 pharmaceuticals-16-00635-f001:**
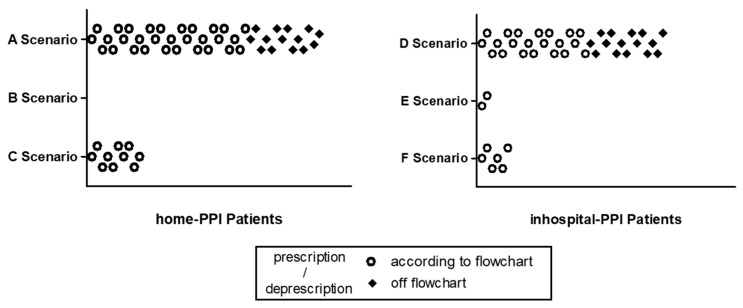
Distribution of patients (n. 98) prescriptions according to scenarios A–F (see Table 2 for details on prescribing scenarios A–F).

**Table 1 pharmaceuticals-16-00635-t001:** Demographic and Prescriptive Characteristics of Patients.

	Total Population (n = 98)		
Demographic Characteristics	No.	%	
Sex			
Male	49	50	
Female	49	50	
Age, years			
Mean ± SD	75.6 ± 10.6		
Polypharmacy (≥5 drugs)	78	79.6	
PPI Prescription Trends	No.	%	Full Dose Prescriptions
home-PPI	54	55.1	83.3%
inhospital-PPI	44	44.9	93.2%
** 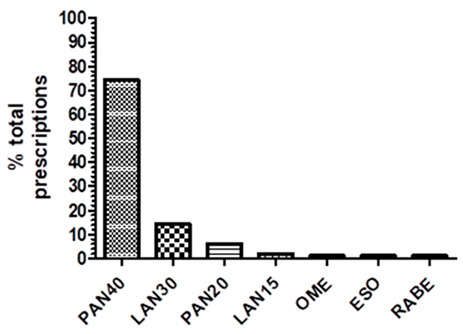 **			PAN40: pantoprazole 40 mg LAN30: lansoprazole 30 mg PAN20: pantoprazole 20 mg LAN15: lansoprazole 15 mg OME: omeprazole any strength ESO: esomeprazole any strength RABE: rabeprazole any strength

Data are expressed as an absolute number (No.), percentage (%), mean ± standard deviation (SD). Home-PPI: PPI prescribed before entering the ward; Inhospital-PPI: PPI prescribed during hospitalization.

## Data Availability

Data is contained within the article.

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
