# Peer review of "Prescriptive Appropriateness: Inhospital Adherence to Proton Pump Inhibitors Deprescription Flow Chart"

_pharmaceuticals, 2023, doi:10.3390/ph16050635_

Round 1

Reviewer 1 Report

This manuscript tackles the debated topic of proton pump Inhibitors (PPIs) prescriptive appropriateness. PPIs are often misused, and their over-prescription is a topic discussed from different viewpoints by many literature sources. The authors focalized on evaluating the efficacy of implementing a pre-elaborated flowchart with the objective of facilitating PPI deprescribing in a setting of ordinary clinical practice. The provided data are of sure interest; however, some aspects of the paper can be improved, and some clarifications are necessary.

Introduction

Line 49: I would suggest adding context to these statements (i.e. literature sources state that…)  as the references provided are heterogeneous both in terms of the national health systems analysed and the time in which the analyses were conducted. Thus, they might not represent the current reality of the clinical practice setting. 

Line 84: I suggest at least acknowledging the presence of Real-world studies relative to the safety and tolerability of PPIs.  

Results

Line 102: The addition of data regarding the pathologies for which patients are treated with a PPI could provide more context to the exposed results.

Line 103: In my opinion, being that 79.6% of patients were treated with more than 5 drugs, including data relative to the drugs administered concomitantly to the PPIs (maybe in table form) could be of great interest.

Material and methods

Line 263: The modified flowchart used for the study was not provided as review material, its inclusion as a figure could significantly improve the readability of the manuscript.

Line 301: could the authors please specify what they intend for “prescription strength”? did they mean appropriateness/adherence to label indications?

Reviewer 2 Report

Thank you for the opportunity to review this interesting paper, which reports the results of a prospective study applying an innovative tool for the deprescription of PPIs. I found the article to be clearly written and well structured. The Introduction is informative, the Methods are described in details (although they have been wrongly placed but can be fixed easily), the Results are clearly presented and well discussed in Discussion. Minor spell check of language required.

Reviewer 3 Report

Introduction can be concise with relevant rationale of the study. Unnecessary content can be trimmed out.

Methodology can be further explained.

Results look descriptive and sound.

The discussion can be improved by providing the detailed rationale of study findings 

Reviewer 4 Report

Line 40, consider splitting the sentence into two. 

Line 79 write abbreviation in full when mentioning it first in text

In abstract you mention . PPIs are the most overprescribed drugs often without considering the benefit/risk ratio - this seems like quite bold statement. Unless there is a specific reference for this (no need to cyte it in abstract) please rephrase.

Readers would benefit from knowing what is the status of IPP drugs in italy, are some OTC? Also you could mention potential of interactions of these drugs and how they may alter absorption of some commonly used drugs.

Table 1 and Figure 1 should be split ... also, define abbreviations under the figure.

What was the sampling method for patients and why were not all included in analysis?

It would be quite interesting to conduct this research with a control group.
